# G Protein-Coupled Receptor Genes, PTGDR1, PTGDR2, and PTGIR, Are Candidate Epigenetic Biomarkers and Predictors for Treated Patients with HPV-Associated Oropharyngeal Cancer

**DOI:** 10.3390/microorganisms8101504

**Published:** 2020-09-29

**Authors:** Kiyoshi Misawa, Atsushi Imai, Takeharu Kanazawa, Masato Mima, Satoshi Yamada, Daiki Mochizuki, Taiki Yamada, Daichi Shinmura, Ryuji Ishikawa, Jyunya Kita, Yuki Yamaguchi, Yuki Misawa, Hiroyuki Mineta

**Affiliations:** 1Department of Otolaryngology/Head and Neck Surgery, Hamamatsu University School of Medicine, Hamamatsu 431-3192, Japan; imaimimi@yahoo.co.jp (A.I.); tendoon@gmail.com (M.M.); veridique.star@gmail.com (S.Y.); daiki_m525@yahoo.co.jp (D.M.); rubybrown4026@yahoo.co.jp (T.Y.); arbestel.84@gmail.com (D.S.); ryu710ryu@gmail.com (R.I.); ta6e9oro.junya@gmail.com (J.K.); mamezou230814@yahoo.co.jp (Y.Y.); mswyuki@abox3.so-net.ne.jp (Y.M.); mineta@hama-med.ac.jp (H.M.); 2Department of Otorhinolaryngology/Head and Neck Surgery, Jichi Medical University, Shimotsuke, Tochigi 329-0498, Japan; kanatake@omiya.jichi.ac.jp

**Keywords:** HPV, oropharyngeal cancer, liquid biopsy, circulating tumor DNA, G protein-coupled receptor, epigenetic markers

## Abstract

Differences in the biology of human papillomavirus (HPV)-associated oropharyngeal cancers (OPCs) and HPV-negative OPCs may have implications in patient management. Early detection is imperative to reduce HPV-associated OPC mortality. Circulating tumor DNA (ctDNA) can potentially serve as a biomarker for monitoring clinically relevant cancer-related genetic and epigenetic modifications. We analyzed the methylation status of 24 G protein-coupled receptor (GPCR) genes in verification (85 OPC primary samples) and validation (8 OPC ctDNA samples) studies using quantitative methylation-specific polymerase chain reaction (Q-MSP). The Q-MSP-based verification study with 85 OPC primary samples revealed the GPCR genes that were significantly associated with recurrence in high methylation groups (≥14 methylated genes) with OPC and HPV-associated OPC (*p* < 0.001). In the Kaplan–Meier estimate and multivariate Cox proportional hazard analyses, 13 GPCR genes were significantly related to increased recurrence in the methylation group. Furthermore, the validation study on ctDNA showed that three of these genes (Prostaglandin D2 receptor 1: *PTGDR1*, Prostaglandin D2 receptor 2: *PTGDR2*, and Prostaglandin I2 Receptor: *PTGIR*) had a prediction performance as emerging biomarkers. We characterized the relationship between the methylation status of GPCR genes and outcomes in HPV-associated OPC. Our results highlight the potential utility of ctDNA methylation-based detection for the clinical management of HPV-associated OPC.

## 1. Introduction

G protein-coupled receptors (GPCRs) are the largest class of cell-surface receptors involved in several cancers, including head and neck squamous cell carcinoma (HNSCC) [1]. Studies have revealed the aberrant expression of GPCRs in human cancers in response to unfavorable events such as gene point mutations, changes in gene copy number, and gene silencing via epigenetic modifications [2]. Recent studies have demonstrated the important contribution of GPCRs to several facets of tumorigenesis, including proliferation, survival, angiogenesis, invasion, metastasis, and therapy resistance [3]. Therefore, detection of aberrant circulating tumor DNA (ctDNA) specific for GPCRs may potentially serve as a minimally invasive monitoring strategy for the early detection of tumor recurrence.

HNSCC prognosis remains poor, given its high recurrence rate [4]. Some clinicopathological parameters, including primary site, nodal involvement, tumor thickness, and status of surgical margins, have been associated with prognosis [5]. Current diagnostic strategies for post-treatment surveillance of patients with HNSCC involve monitoring by clinical evaluation in combination with flexible endoscopy and conventional imaging [6]. Advances in genomic technology have provided vast information on different cancer subtypes and introduced new therapeutic targets [7]. ctDNA may potentially be used to monitor early disease relapse and offer a dynamic molecular reflection of genomic alterations in cancer. 

Human papillomavirus (HPV) is known to cause cancers of the cervix, vulva, vagina, penis, anus, and oropharynx. Cervical cancer is the most common HPV-associated cancer among women, and oropharyngeal cancers (OPC) are the most common among men [8]. It is characterized with a few genomic alterations in *TP53* and *p16* but may also exhibit phosphatidylinositol-4,5-bisphosphate 3-kinase catalytic subunit alpha (*PIK3CA*) alterations, copy-number gains in TNF receptor-associated factor 3 and E2F transcription factor 1 (E2F1), and lack of cyclin D1 (*CCND1*) amplification as compared to HPV-negative cancers [9]. Studies have revealed the correlation between DNA methylation and several cancer types, especially virus-associated cancers [10]. Researchers have performed DNA methylation analysis to further subdivide patients with HPV-associated and HPV-negative OPC, thereby successfully recapitulating the distinct prognosis [11,12,13]. 

To our knowledge, our study is the first to suggest that prostaglandin D2 receptor type 1 and 2 (*PTGDR1* and *PTGDR2*) and prostaglandin I2 receptor (*PTGIR*) methylation is associated with worse disease-free survival (DFS) and serves as a critical event in HPV-associated OPC. Here, we report the results of a combined verification and validation study performed to identify novel ctDNA-based methylation markers and document their ability to efficiently improve the diagnosis and prognosis of HPV-associated OPC patients with HPV-ctDNA-negative.

## 2. Materials and Methods

### 2.1. Clinical Tumor Samples 

Oropharyngeal tumor samples were obtained from 85 patients who underwent treatment at the Department of Otolaryngology/Head and Neck Surgery, Hamamatsu University School of Medicine (Hamamatsu, Shizuoka, Japan). Written informed consents were obtained from all patients before treatment, and the experimental protocol was approved by the Hamamatsu University School of Medicine (ethics code: 25-149 and 17-041). The ratio of male to female was 65:20, and the mean age of the patients was 63.4 years (range, 39 years to 85 years). Patient characteristics have been shown in Appendix A.

### 2.2. Liquid Biopsy

We isolated ctDNA from 4.0 mL plasma samples by affinity-based binding to magnetic beads as per the manufacturer’s instructions (QIAamp MinElute ccfDNA Kit, QIAGEN, Hilden, Germany). Peripheral blood samples (10 mL each) were collected in cell-stabilizing tubes (Cell-Free DNA Collection Tube, Roche, CA, USA). Eight patients with HPV-associated OPC were enrolled for validation analysis. Patient characteristics have been shown in Appendix A.

### 2.3. Detection of High-Risk HPV DNA Using PCR and p16 Immunostaining from Tumor Tissue Samples

For high-risk HPV DNA detection, samples were assessed by PCR using specific primers for HPV types 16, 18, 31, 33, 35, 52, and 58. The prevalence of HPV DNA was analyzed with the PCR HPV Typing Set (TaKaRa, Tokyo, Japan). The PCR products were separated by electrophoresis on 9% polyacrylamide gels followed by staining with 0.5 g/mol ethidium bromide. For the immunohistochemistry analysis, an anti-human p16 monoclonal antibody (clone: E6H4, Roche Diagnostics GmbH, Germany) was used.

### 2.4. Bisulfite Modification and Quantitative Methylation-Specific Polymerase Chain Reaction (Q-MSP)

DNA extraction from fresh tissues was performed using a QIAamp DNA Mini Kit (QIAGEN, Hilden, Germany). Sodium bisulfite conversion was performed using the MethylEasy Xceed Rapid DNA Bisulfite Modification Kit (TaKaRa) as per the manufacturer’s instructions. Aberrant DNA-methylation, often reported around the transcription start site (TSS) within a CpG island, was evaluated by Q-MSP. The primer sequences specific for G protein-coupled receptor (GPCR) genes have been shown in Appendix A. A standard curve for Q-MSP was constructed by plotting five serially diluted standard solutions of EpiScope Methylated HeLa gDNA (TaKaRa). Normalized methylation value (NMV) for PCR conditions was analyzed as previously described [10,14]. A standard curve was generated from serial dilutions of EpiScope Methylated HeLa gDNA (TaKaRa). NMV was defined as follows: NMV = (GPCR-S/GPCR-FM)/(ACTB-S/ACTB-FM), where GPCR-S and GPCR-FM represent GPCR methylation level in sample and universally methylated DNA respectively, and ACTB-S and ACTB-FM correspond to β-actin level in sample and universally methylated DNA, respectively.

### 2.5. Detection of HPV-16 DNA by Quantitative PCR from Liquid Biopsy Samples

To identify HPV-16 genomes, samples were subjected to quantitative PCR using specific primers for HPV type 16. The primers used for HPV-16 E7 sequence detection were as follows: 5′-TCCAGCTGGACAAGCAGAAC-3′ (forward primer) and 5′-CACAACCGAAGCGTAGAGTC-3′ (reverse primer) [11].

### 2.6. Data Mining in the Cancer Genome Atlas (TCGA)

The MethHC (http://methhc.mbc.nctu.edu.tw/php/index.php) was used to extract data from TCGA (available in August 2019). DNA methylation of GPCRs genes was measured by Illumina Infinium Human Methylation 450 K BeadChip. The methylation score for each CpG site is represented as β values and ranges from 0 to 1, corresponding to unmethylated and completely methylated DNA, respectively. In addition, RNA Sequencing (RNAseq) data were obtained from the TCGA data portal (https://tcga-data.nci.nih.gov/tcga/).

### 2.7. Data Analysis and Statistics

A receiver operator characteristic (ROC) curve analysis of target genes was performed using the NMVs for 36 matched-paired HNSCC and normal mucosal samples on the Stata/SE 13.0 system (Stata Corporation, College Station, TX, USA). To determine the area under the ROC curve, true positive rate (sensitivity) was plotted as a function of false positive rate (1-specificity) for different cut-off points and NMV threshold was calculated for each target gene. Cut-off values corresponding to the highest accuracy were determined based on sensitivity/specificity, as indicated in Appendix A. Methylation index was defined as the number of genes with promoter methylation [15]. A Student’s *t*-test was performed to evaluate the association between clinical variables and methylation index. DFS was investigated using the Kaplan–Meier method and the log-rank test. The probability of survival was evaluated by generating a Kaplan–Meier curve. Cox’s proportional hazard regression analysis that included age (≥ 0 years vs. <70 years), sex, alcohol intake, smoking status, and tumor stage (I vs. II–IV), and methylation status was used to identify multivariate predictive values of prognostic factors. A value of *p* < 0.05 was considered statistically significant.

## 3. Results

### 3.1. Verification Analysis of the Methylation Status in OPC Tissue Samples

Q-MSP analysis of the methylation status of 24 GPCR genes was performed using 85 primary OPC samples, including 48 HPV-associated OPC and 37 HPV-negative OPC samples (Figure 1A). Methylation frequencies for these genes have been summarized in Figure 1B. Considering the mean methylation index values, no significant association was observed between HPV-associated OPC and HPV-negative OPC groups (Figure 1C).

### 3.2. Association between Methylation Index and Clinicopathological Characteristics

Methylation index was defined as the ratio between the number of methylated genes and total number of tested genes in each sample. Mean differences in methylation index based on age at onset, sex, smoking habit, alcohol consumption, tumor size, lymph node status, clinical stage, and recurrence have been illustrated in Figure 2. The methylation index was significantly higher in smokers (10.32 ± 5.48) and recurrence-positive cases (12.60 ± 5.62) than in non-smokers (7.73 ± 3.57, *p* = 0.030) and recurrence-negative cases (8.58 ± 4.57, *p* = 0.002) (Figure 2A). Among HPV-associated OPC cases, the methylation indices in smokers (11.31 ± 5.36) and recurrence-positive cases (13.11 ± 4.37) were significantly higher than those in non-smokers (7.63 ± 3.48, *p* = 0.011) and recurrence-negative cases (9.10 ± 4.89, *p* = 0.028) (Figure 2B). Moreover, the methylation index was significantly higher in recurrence-positive cases (12.18 ± 6.66) than in recurrence-negative cases (7.81 ± 4.02, *p* = 0.018) among patients with HPV-negative OPC (Figure 2C). 

### 3.3. Kaplan–Meier Estimates in 85 Patients with OPC

Based on the results of the log-rank test, DFS for the group with ≥14 methylated genes was lower than that for the group with <14 methylated genes (14.2% vs. 84.0%; log-rank test, *p* = 0.0003; Appendix A). We found an association between poor survival and methylation phenotype defined as ≥14 methylated genes (*p* = 0.0003; Figure 3A). DFS did not significantly differ between patients with HPV-associated OPC and those with HPV-negative OPC (Figure 3B). In 48 HPV-associated patients with OPC, DFS was lower in patients with ≥14 methylated genes than in those with <14 methylated genes (31.8% vs. 87.8%, respectively; *p* = 0.0056; Figure 3C). Among the 37 patients with HPV-negative OPC, DFS was 0.0% in cases with ≥14 methylated genes and 78.6% in cases with <14 methylated genes (log-rank test, *p* = 0.045) (Figure 3D).

### 3.4. Kaplan–Meier Estimates for Patients with HPV-Associated OPC

The Kaplan–Meier survival curves for each of the 24 genes in patients with HPV-associated OPC are shown in Figure 4. No significant difference was observed in DFS between patients with methylated genes and those with unmethylated genes. A notable exception was that DFS was significantly shorter when Galanin receptor 2: *GALR2*, Growth hormone secretagogue receptor: *GHSR*, Neuromedin U Receptor 1: *NMUR1*, Neuropeptide Y Receptor Y1: *NPY1R*, Neuropeptide Y Receptor Y2: *NPY2R*, Neuropeptide Y Receptor Y5: *NPY5R*, Neurotensin receptor 2: *NTSR2*, *PTGDR1*, and *PTGDR2* were methylated (*p* = 0.001, *p* = 0.001, *p* = 0.001, *p* = 0.028, *p* = 0.026, *p* = 0.010, *p* = 0.012, *p* = 0.007, and *p* = 0.004, respectively). 

### 3.5. Stratification Analysis 

The relationship between methylation status and recurrence risk was analyzed by a multivariate analysis using a Cox proportional hazard regression model adjusted for age, sex, smoking status, alcohol consumption, and clinical stage. In patients with OPC (full panel), the hypermethylation of *GALR2*, *GHSR*, *NMUR1*, *NPY4R*, *PTGDR2*, *PTGER4*, *PTGIR*, and *TBXA2R* was associated with significantly reduced DFS at a hazard ratio of 4.509 (95% confidence interval (CI): 1.560–13.03), 5.225 (95% CI: 1.540–17.73), 5.589 (95% CI: 1.582–19.74), 3.859 (95% CI: 1.320–11.28), 5.165 (95% CI: 1.513–17.63), 5.312 (95% CI: 1.036–27.23), 3.617 (95% CI: 1.267–10.32), and 3.963 (95% CI: 1.163–13.50), respectively (Figure 5A). In patients with HPV-associated OPC, the hypermethylation of *GALR2*, *GHSR*, and *NMUR1* was related to significantly reduced survival at a hazard ratio of 23.62 (95% CI: 2.453–227.5), 31.18 (95% CI: 2.697–360.4), and 9.909 (95% CI: 1.502–65.39), respectively (Figure 5B). In patients with HPV-negative OPC, the methylation status of *TACR1* promoter negatively correlated with recurrence (odds ratio (OR), 0.059; 95% CI: 0.007–0.533) (Figure 5C). 

A prognostic risk category based on the methylation statuses of *GALR2*, *GHSR*, *NMUR1*, *NPY1R*, *NPY2R*, *NPY4R*, *NPY5R*, *NTSR2*, *PTGDR1*, *PTGDR2*, *PTGER4*, *PTGIR*, and *TBXA2R* refined the risk stratification for outcomes as an independent prognostic factor for HPV-associated OPC (Appendix A).

### 3.6. Validation Analysis of Methylation Status in Clinical Primary Samples and Paired ctDNA Samples

We found a concordance between the primary samples and matched-pairs before treatment ctDNA in HPV-associated OPC for *PTGDR1*, *PTGDR2*, and *PTGIR* promoter methylation but not for *GALR2*, *GHSR*, *NMUR1*, *NPY1R*, *NPY2R*, *NPY4R*, *NTSR2*, *NPY5R*, *PTGER4*, and *TBXA2R* promoter methylation (Figure 6A,B). In HPV-associated OPC patients with HPV-ctDNA-negative (Case 2 and Case 3), *PTGDR1*, *PTGDR2*, and *PTGIR* promoter methylation signals were positive (Figure 6B). We further compared methylation of these 13 markers in eight matched-paired post-treatment ctDNA and found that the methylation signals of *PTGDR1*, *PTGDR2*, and *PTGIR* were disappeared (Figure 6C). We tested these three genes in the validation study, which could distinguish pre-treatment ctDNA from post-treatment ctDNA. 

### 3.7. Analysis of Methylation and Expression Data from TCGA

The methylation status of 24 GPCRs gene promoters was estimated in an additional 516 HNSCC samples and 50 normal samples from TCGA. The average β-values (indicating promoter methylation) for the GPCRs genes were significantly higher in the HNSCC samples than in the normal samples (*p* < 0.05), except for *PTGER4* and *TBXA2R*. Moreover, we have compared the expression level of the GPCRs genes between HNSCC and normal control using the samples from TCGA database (Appendix A).

## 4. Discussion

HPV-associated OPC initially arises in tonsillar crypts, explaining the high prevalence of HPV in the tonsils [16]. Patients present for medical evaluation only at the advanced stage of OPC, given the lack of an effective screening program due to the absence of identified precursor lesions [17]. Meanwhile, the incidence and mortality of cervical cancer have decreased [18]. Screening methods now include tests for high-risk strains of HPV central to the pathogenesis of cervical cancer [19]. Early detection will increase the chances of successful treatment and functional preservation. Our study demonstrates the potential usefulness of the assessment of ctDNA-based methylation status for the clinical diagnosis and treatment monitoring of HPV-associated OPC.

HPV is now established as the principal cause of increased incidence of an HNSCC subset in numerous geographic regions around the world [20]. HPV-associated OPC typically presents in a younger, healthier population with a different set of risk factors and good prognosis for survival [21]. However, several studies have demonstrated the risk stratification of patients with HPV-associated OPC based on tobacco smoking history as follows: low risk (HPV-associated, 10 pack years) and intermediate risk (HPV-associated, >10 pack years) [22,23]. In comparison to patients with non-metastatic OPC, those with a higher rate of distant metastatic disease present smoking history, age ≥ 50 years, N3 disease, T4 disease, and p16-negative disease [24]. Despite these acknowledged limitations, it is imperative to investigate the specific patient population at the highest risk.

The epigenetic fingerprint of HPV-associated HNSCC is very different from that of HPV-negative cancers [25]. HPV E6 and E7 have been found to be directly associated with the activity of DNA methyltransferases (DNMTs) [26]. These epigenetic alterations are often detected early during tumorigenesis and are likely to be the key initiating events in HPV-associated cancers [27]. The differences in the methylation statuses of three genes, *PAX1*, *SOX1*, and *ZNF582,* were evident between normal control samples and cervical intraepithelial neoplasm (CIN) I–III as well as SCC samples [28,29].

The prevalence of high-risk HPV in nonmalignant tonsils is low. According to PCR, 2 of the 195 (1.0%) DNA samples isolated from the tonsil tissue were positive for HPV-16 in a German group study [30]. At the Helsinki University Hospital, 13 of the 206 (6.3%) samples were positive for HPV DNA in a 2001 to 2003 study and 5 of the 477 (1.0%) patients were detected positive for HPV DNA in a 2012 and 2015 study [31,32]. The observation that oral rinse test is less sensitive in the detection of HPV-associated tonsil cancers highlights its limitation as a screening tool. Of the 81 normal cohort patients with either thyroid cancer or benign thyroid nodules, 79 (97.5%) had no HPV in their oral rinse in the PCR test and two patients were positive for HPV-16; however, clinical examination showed no clinical evidence for OPC or oral cavity cancer [33]. Based on p16 immunostaining, the sensitivity and specificity of HPV-DNA detection in oral rinse were 75% and 100%, respectively [34]. Thus, additional approaches are warranted to improve oral HPV test characteristics for the early detection of OPC.

Biomarkers have the potential to meet one of the most important unmet clinical needs in OPC, a risk classification system that is more effective than medical history and physical examination [35]. Patients with undetectable HPV-ctDNA during clinical follow-up are unlikely to have recurrent disease and may be spared from routine radiographic and in-office pharyngoscopy surveillance [36]. HPV-ctDNA detection may also confirm recurrent disease when tissue biopsy is deemed high risk [22]. Implementation of liquid biopsy should be further evaluated on the basis of methodological aspects and cost effectiveness [37]. At present, only a few studies have investigated the relationship between ctDNA methylation events and prognosis in HNSCC [38,39,40]. Recently, de Jesus et al. reported that methylated ctDNA was detected in 11/15 (73.3%) of these samples: 7/11 with methylation in *CCNA1*, 2/11 with methylation in *TIMP3*, 1 case with methylation in *CDH8*, and 1 in *DAPK* [38]. Currently, there is a lack of an effective OPC screening program because there are no identified OPC precursor lesions. In the future, we think that highly sensitive and tissue-specific ctDNA biomarkers of HPV-associated OPC will be discovered, allowing early detection. The current study is the first to examine GPCRs genes’ methylation levels in HPV-associated OPC. Additionally, our ctDNA findings showed that *PTGDR1*, *PTGDR2*, and *PTGIR* methylation levels could aid in real-time disease surveillance and serve as an alternative method of screening for HPV-associated OPC patients with HPV-ctDNA-negative.

## 5. Conclusions

In summary, the results of the current study demonstrate three new promising methylation markers (*PTGDR1*, *PTGDR2*, and *PTGIR*) related to HPV-associated OPC from an unbiased comprehensive stepwise verification and validation study. Our findings implicate analyses using DNA methylation-based strategies for the detection of ctDNA and suggest that such approaches may serve as a strategy for early cancer diagnosis and therapeutic assessment.

## Figures and Tables

**Figure 1 microorganisms-08-01504-f001:**
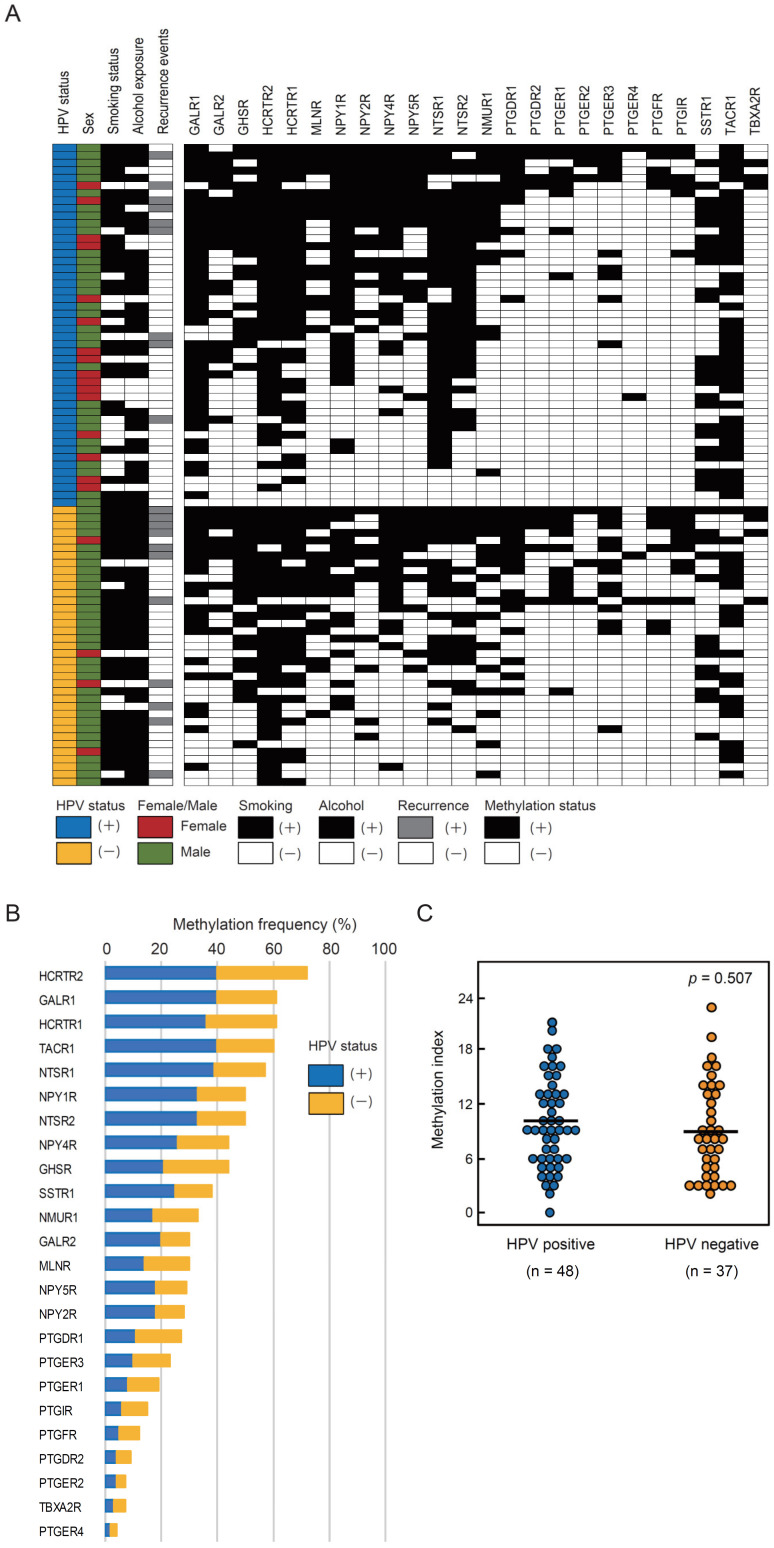
Methylation frequencies for 24 G protein-coupled receptor (*GPCR)* genes in oropharyngeal cancer (OPC) tissue samples. (**A**) Comparison of methylation statuses of the promoters of 24 *GPCR* genes in patients with human papillomavirus (HPV)-associated and HPV-negative OPC. (**B**) Bar graph showing methylation frequencies (%) of 24 *GPCR* genes in the verification cohort. Blue bars, samples of HPV-associated cases; orange bars, samples of HPV-negative cases. (**C**) The mean methylation index values for HPV-associated and HPV-negative groups have been compared using the Student’s t-test.

**Figure 2 microorganisms-08-01504-f002:**
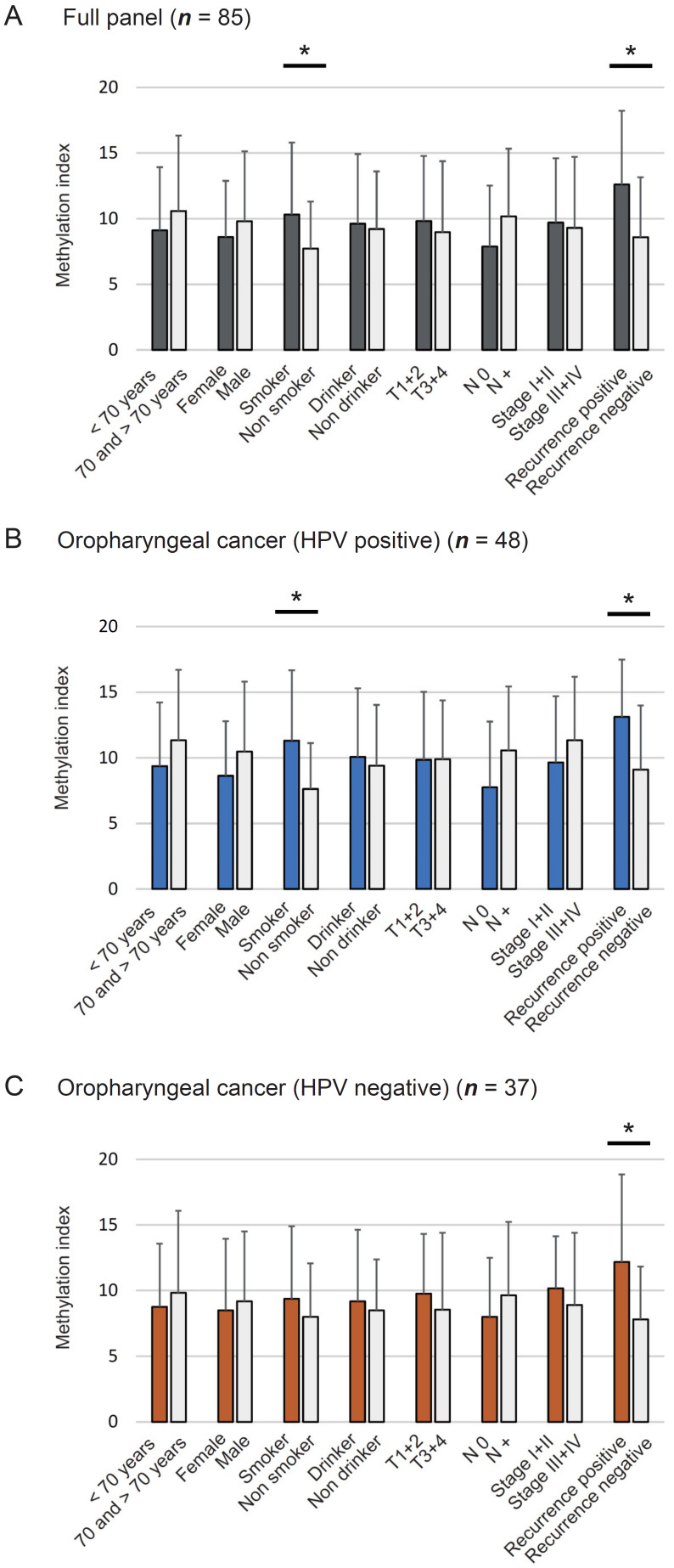
Association between methylation index and selected clinical parameters. The mean methylation index values for various groups have been compared using the Student’s t-test to detect any association between methylation index and selected epidemiologic and clinical characteristics: (**A**) Eighty-five full panel cases. No differences have been noted with respect to clinical characteristics, (**B**) HPV-associated OPC, (**C**) HPV-negative OPC. Mean and standard deviation (SD) have been also indicated, and statistical comparisons between groups are depicted. * *p* < 0.05 indicates the statistical significance.

**Figure 3 microorganisms-08-01504-f003:**
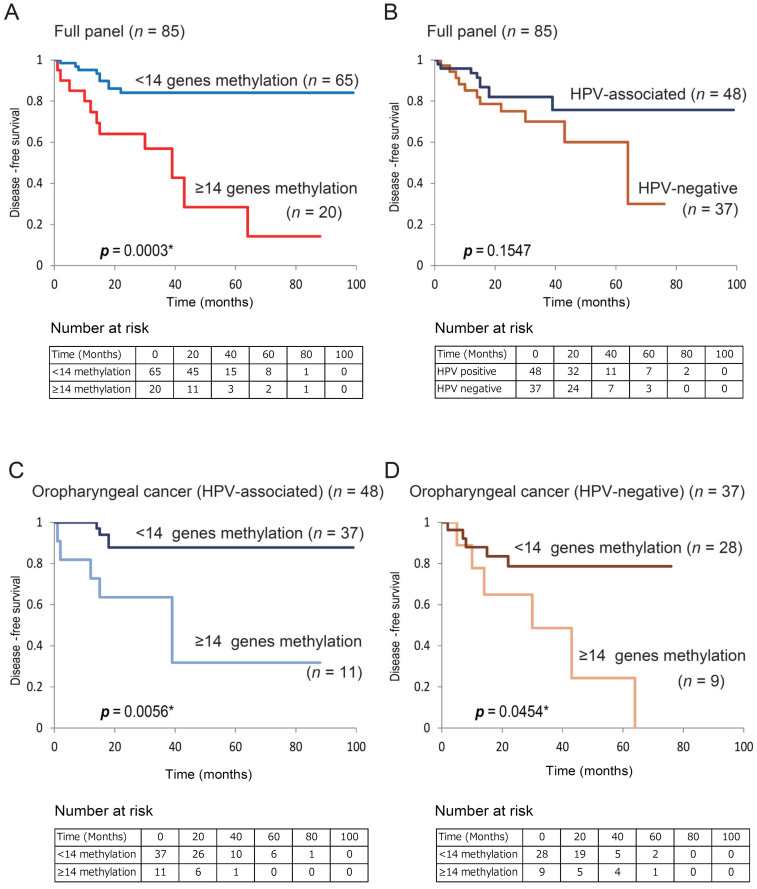
Kaplan–Meier survival curves for patients with OPC based on GPCR gene methylation status. Disease-free survival based on (**A**) methylation status in full cases (*n* = 85), (**B**) HPV status in full cases (*n* = 85), (**C**) methylation status in HPV-associated OPC cases (*n* = 48), (**D**) methylation status in HPV-negative OPC cases (*n* = 37). **p* < 0.05.

**Figure 4 microorganisms-08-01504-f004:**
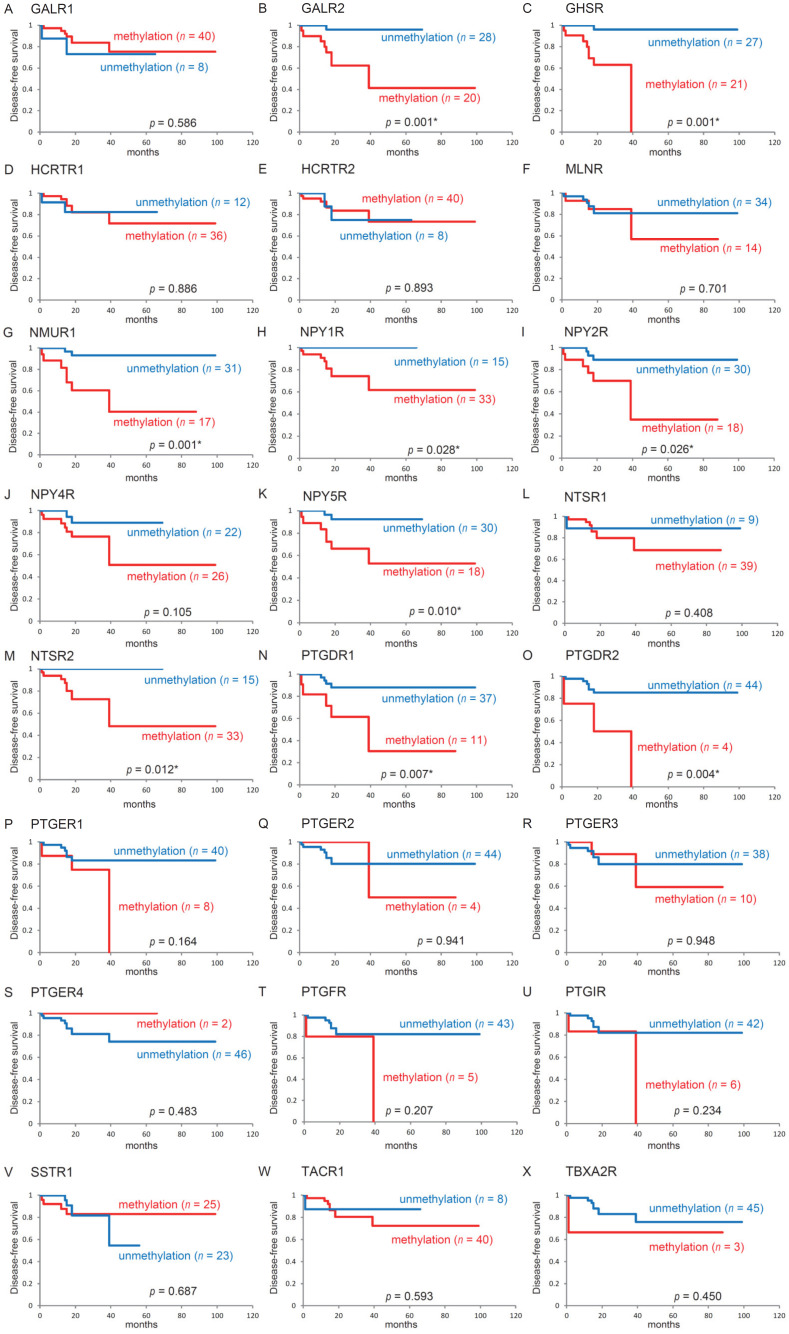
Kaplan–Meier survival curves for 48 patients with HPV-associated oropharyngeal cancer according to the methylation statuses of 24 target genes disease free survival (DFS) for (**A**) *GALR1*, (**B**) *GALR2*, (**C**) *GHSR*, (**D**) *HCRTR1*, (**E**) *HCRTR2*, (**F**) *MLNR*, (**G**) *NMUR1*, (**H**) *NPY1R*, (**I**) *NPY2R*, (**J**) *NPY4R*, (**K**) *NPY5R*, (**L**) *NTSR1*, (**M**) *NTSR2*, (**N**) *PTGDR1*, (**O**) *PTGDR2*, (**P**) *PTGER1*, (**Q**) *PTGER2*, (**R**) *PTGER3*, (**S**) *PTGER4*, (**T**) *PTGFR*, (**U**) *PTGIR*, (**V**) *SSTR1*, (**W**) *TACR1*, and (**X**) *TBXA2*R for methylated (red lines) and unmethylated (blue lines) genes. A value of *p* < 0.05 has been considered statistically significant.

**Figure 5 microorganisms-08-01504-f005:**
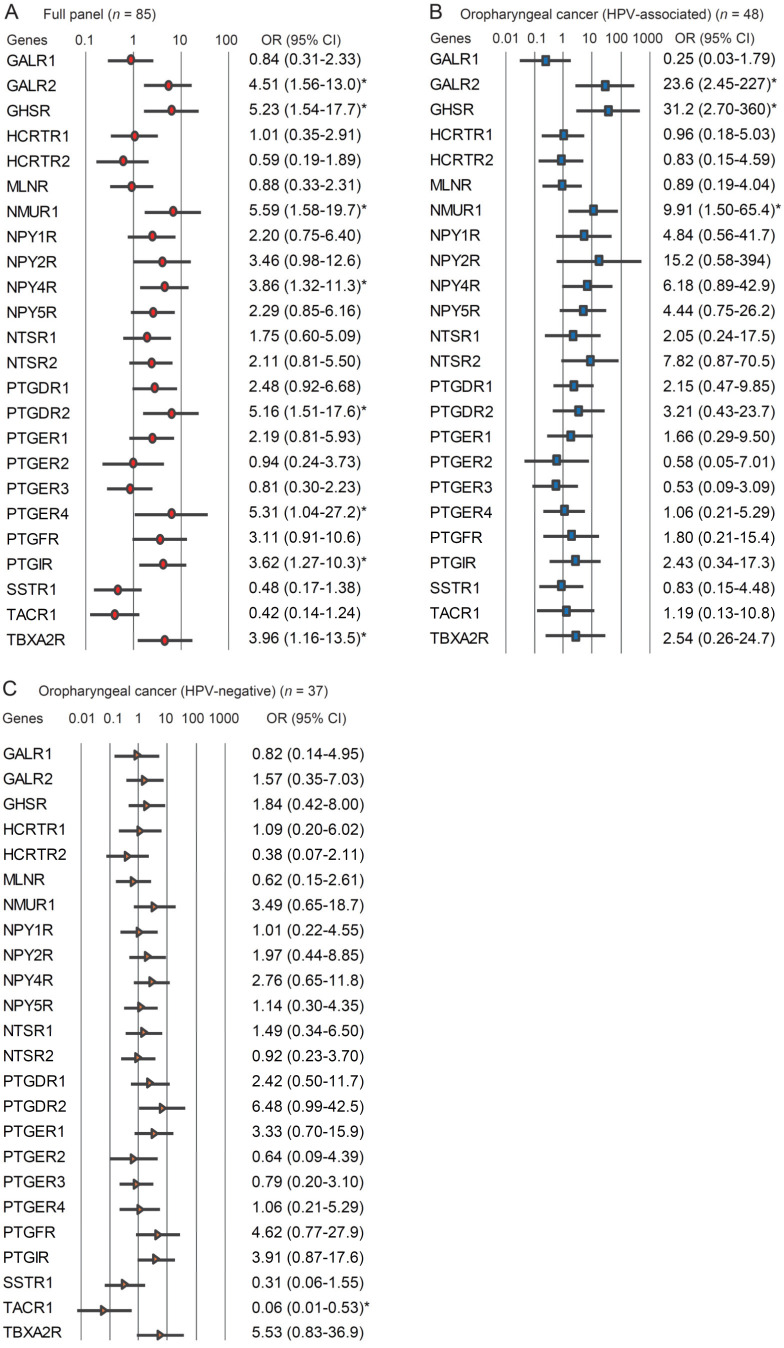
Risk of recurrence based on 24 *GPCR* genes’ odds ratios (OR) for recurrence were determined using a Cox proportional hazard model adjusted for age (≥70 vs. <70 years), sex, alcohol intake, smoking status, and tumor stage (I vs. II–IV). (**A**) Full panel of OPC (*n* = 85), (**B**) HPV-associated OPC (*n* = 48), (**C**) HPV-negative OPC (*n* = 37). OR: odds ratio. CI: confidence interval. * *p* < 0.05.

**Figure 6 microorganisms-08-01504-f006:**
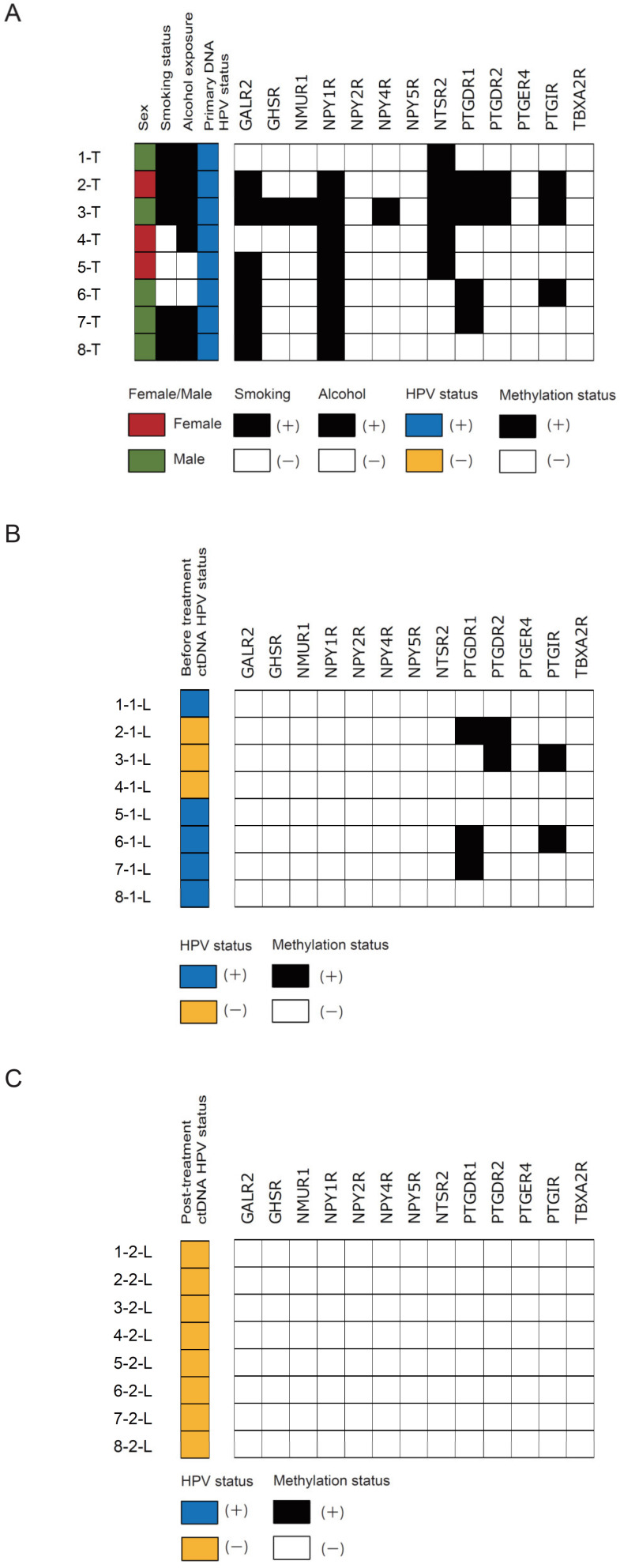
Validation analysis of methylation status in clinical primary samples and paired ctDNA samples. Comparison of methylation statuses represents the methylation status of these 13 markers in (**A**) 8 primary samples, (**B**) ctDNA isolated from the same patients before treatment as above, and (**C**) post-treatment.

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
