# Peer review of "G Protein-Coupled Receptor Genes, PTGDR1, PTGDR2, and PTGIR, Are Candidate Epigenetic Biomarkers and Predictors for Treated Patients with HPV-Associated Oropharyngeal Cancer"

_microorganisms, 2020, doi:10.3390/microorganisms8101504_

Round 1

Reviewer 1 Report

Misawa et al. investigated the methylation pattern status of 24 GPCR in primary tumors as well as ctDNA of oropharyngeal cancers (PC) with and without HPV. Their results showed that high methylation in some of GPCRs is associated with recurrence regardless of HPV status. Validation study on ctDNA suggested that three of those genes, PTGDR1, PTGDR2 and PTGIR could be useful as a biomarker predicting recurrence of HPV positive OPC. This is well organized study with interesting results. Good liquid biopsy markers for OPC could significantly improve prognosis of OPC. Thus study could be valuable for OPC diagnosis in the future. I believe this article is suitable for publication. However, a few points need to be addressed.

Major points

1. In materials and methods, 2.4.

It can be read as if only HPV16 was tested to determine HPV status in specimen. Other high risk HPV types have been detected in OPC. All high risk HPV should be tested, even though the likelihood of high risk HPV other than HPV16 is very small. Did you validate the absence of HPV in other way for OPC with no HPV16 DNA, like maybe p16 staining?

 Minor points

2. Authors offered no explanation for why these 24 GPCR are selected. Since there are more than 300 GPCRs, it would be helpful to explain why these 24 genes were picked up.

3. In the same regard as (2), the promoter methylation typically results in reduced gene expression. Is it known that those 24 GPCRs expression are reduced in OPC? I am interested since some GPCRs are known to be upregulated in cancers.

4. In introduction, line 53~, OPC is NOT the most common HPV-associated cancers worldwide. The statement is true in some countries such as US due to successful implementation of HPV vaccine and early detection. However, in majority of the countries, cervical cancer is still the most common HPV associated cancer. The statement needs to be corrected.

Author Response

September 25, 2020

Prof. Tomokazu Yoshizaki

Special Issue Editor

Microorganisms: Special Issue " Pathogenic Role of Virus Infection in Head and Neck Tumors"

Dear Editor:

We appreciate the opportunity to revise our manuscript entitled “G protein-coupled receptor genes, PTGDR1, PTGDR2, and PTGIR, are candidate epigenetic biomarkers and predictors for treated patients with HPV-associated oropharyngeal cancer.” (manuscript ID, microorganisms-929233). Please refer to our point-by-point responses to the reviewers’ comments and questions below. All changes made to the revised manuscript are highlighted in red. We hope that our manuscript will now be deemed suitable for publication in Microorganisms.

Sincerely,

Kiyoshi Misawa

Department of Otolaryngology/Head and Neck Surgery

Hamamatsu University School of Medicine

1-20-1 Handayama

Shizuoka 431-3192, Japan

Phone: 81-53-435-2252

Fax: 81-53-435-2253

Reviewer 2 Report

  1. Authors have tested only one type of high-risk HPV (HPV-16) even though other types such as HPV – 18, 33, 35 etc. are known to contribute to OPC. Please explain why other genotypes where not tested or how are they certain that HPV negative samples are true negatives?
  2. Authors use different terminology for describing the methylation status. E.g. NMV, methylation frequency, methylation index. How they are interrelated or different? Please explain.
  3. What exactly meant by ‘recurrence’? Is it recurrence of OPC or HPV status?
  4. If recurrence refers to OPC, then what tissues where collected in recurrence negative samples? (Fig. 2A for e.g.)
  5. What kind of treatment(s) where given to patients? This becomes important especially in Fig 6C, that suggests post-treatment all 5 HPV +ve patients could get rid of HPV in their tissue? Was HPV maintained as episomes since integrated HPV is hard to get rid-of? Authors should explain in details.
  6. Discussion section hardly discusses the findings of the current research that authors carried out. Please re-organize this section explaining the merits of the current research in the context of published research.
  7. The statement “We found a concordance between the primary samples and matched paired pre-treatment ctDNA in HPV-associated OPC for PTGDR1, PTGDR2, and PTGIR promoter methylation” (line 198 onwards) is highly misleading since various methylation positive genes had HPV negative samples (6B)? For e.g. PTGDR2 had two HPV -ve (2-1L and 3-1L) samples on pretreatment (6B) that were considered HPV +ve in the Figure 6A. How then they are “HPV associated” OPC? Please clearly explain the results.
  8. Further, the 5 HPV +ve OPC samples (even though authors classify 8 samples as “HPV-associated”, please refer above comment) described for Fig 6 is low hence, I think comparisons should be carried out with a larger sample size.

Minor error that needs attention –

  1. Line 165 – “are shown” (replace has been)
  2. Line 202-203 “matched-paired”
  3. Supplementary information page 1 – flow chart “OPC” meaning?
  4. Supplementary file - What is cfDNA?

And a few other minor errors that authors need to go through thoroughly.

Author Response

(The authors gave the same response as above.)

Round 2

Reviewer 2 Report

Authors have addressed my concerns and incorporated additional information. Track change mode of the PDF makes it a bit difficult to follow, though I think the paper is now acceptable.